# Opportunities to Capitalize on Transylvanian Wood Pastures through Nature-Based Tourism: A Case Study of Viscri Village, Brașov County, Romania

Iuliana Vijulie [1], Mihaela Preda [1,*], Andreea Nita [1,2] and Anca Tudoricu [1]

1   Faculty of Geography, University of Bucharest, 1 Blv. Nicolae Bălcescu, 010041 Bucharest, Romania; iuliana.vijulie@g.unibuc.ro (I.V.); andreea.nita@cc.unibuc.ro (A.N.); anca.tudoricu@g.unibuc.ro (A.T.)
2   Center for Environmental Research and Impact Studies, University of Bucharest, 010041 Bucharest, Romania
*   Correspondence: mihaela.preda@geo.unibuc.ro; Tel.: +40-727-784-038

**Abstract:** European wood-pastures are complex socio-ecological systems, valuable from ecological and cultural perspectives. Over time, they have gone through a decline in area coverage due to overgrazing, abandonment of traditional agricultural practices or the development of intensive agriculture. In Transylvania (Romania), such landscapes are still preserved, but they are very vulnerable. Restoring them and including them in ecotourism circuits could benefit local communities and the environment. In this context, the purpose of the study was to analyze the possibilities for nature-based tourism using the wood-pasture located near Viscri and the respondents' level of awareness of its economic, social, and environmental importance. The main research methods used were GIS and remote sensing techniques, as well as direct field observations and surveys among local farmers and tourists. The study's results pointed out the existence of a community-managed wood-pasture, well preserved through traditional agricultural practices and marked by significant biodiversity. Even though the wood-pasture is not yet attracting strong tourist flows, respondents linked it with various and numerous benefits. The leisure activities that respondents participated in while visiting the wood-pasture generated a very high level of satisfaction, as they declared their intentions to revisit and/or recommend it. Although British King Charles III was the one who carried out the promotion of this wood-pasture due to his particular interest in the area, it was still not enough. Therefore, authorities should acknowledge its value for the development of the local community and be more present in supporting nature-based tourism activities.

**Keywords:** meadows; trees; biodiversity; traditional activities; land use; ecotourism; promotion

## 1. Introduction

### 1.1. Wood-Pasture Coverage in Europe

Wood-pastures in Europe are models/archetypes of High-Nature Value (HNV) farmland, which hold a unique biodiversity value and are socially and culturally important [1–4]. In other words, these are complex socio-ecological systems, products of the long-term interaction between society and the surrounding landscape [5]. Most European wood-pastures were formed in the Middle Ages [6]. Over time, however, their coverage has historically shrunk, either due to the intensification of agricultural activity and changes in land use or due to the abandonment of traditional farming practices within certain European regions—particularly in western and central Europe in the nineteenth century and Mediterranean areas in the second half of the twentieth century [1,7,8].

Wood-pastures account for about 4.7% of the area of EU Member States [1]. They create a landscape that marks the transition from forests to meadows, hosting plant and animal species that can only exist in the mosaic habitats of both forests and meadows [9]. These include pastures with rare trees, pastures in open forests, and pastures hosting cultivated trees (e.g., olive). Pastures with rare trees are extensive, primarily within Mediterranean



countries (e.g., Spain, France, Italy) and Eastern European countries (e.g., Romania, Bulgaria) [1,9]. Pastures located in open forests are mainly concentrated in Spain and Portugal, being called dehesas (Spain) and montados (Portugal) [10–12]. Pastures hosting cultivated trees are found everywhere in the Mediterranean, with significant coverage in Spain, Greece, Portugal and Italy [1,13].

Such wood-pastures are still found in small areas in Germany, where they were very present until the land reforms of the 19th century. Today, they are being reinvigorated to conserve and enhance the landscapes' local biodiversity [14,15]. A path similar to the German pastures has been followed by the wood-pastures in the UK [16]. Hungary faced a decline in traditional farming systems in the 20th century. Only a few wood-pastures are still actively used today, but more and more are being restored and grazed on mainly for conservation purposes [9]. Also, rewilding is a new type of nature management that is being applied in Germany, the Netherlands or the UK to recreate wood-pasture landscapes on abandoned agricultural land with the reintroduction of large herbivores, e.g., Hack cattle and Konik horses, for which grazing refuges have to be created. This type of management aims to restore natural processes with minimal human intervention [17].

Thus, pastures have recently received more attention from researchers in terms of conservation management. Still, more attention is needed to consider the Common Agricultural Policy (CAP), including the EU Rural Development Policy and Habitats Directive. In this respect, claims were made to include unique wood-pastures in EU agricultural policies [1,7].

Wood-pastures are part of Europe's cultural heritage and can contribute significantly to the economic integrity of rural areas, intending to improve ecological quality through sustainable management [7].

In Europe, a few projects are aimed at the restoration and conservation management of wood-pastures, with the most well-known being those from Estonia or Hungary [2,9]. Such financial incentives should be used more widely for the revival of wood-pastures, given their benefits, both in terms of their unique biodiversity and for improving the livelihoods of local people [2,9]. At the same time, there are several opportunities to generate additional income for the local communities that own such wooded grassland landscapes, such as the development of ecotourism or green care services ("a range of activities that promotes physical and mental health and well-being through contact with nature") [18] (p. 120).

In conclusion, these landscapes also provide recreational areas for leisure ecotourism and have an aesthetic value that significantly contributes to developing cultural ecosystem services [19–23]. For example, the Iberian Peninsula's green oak or cork oak pastures (dehesa, montado) are highly valued for their aesthetics [24,25].

### 1.2. Brief History of Wood-Pastures in Transylvania, Romania

Most of the wood-pastures in Transylvania appeared during the Middle Ages and gradually developed over the centuries due to low-intensity environmental activities such as extensive grazing [26,27]. They are mainly located in Saxon villages in the Târnava Mare Region (southern Transylvanian Plateau), where an agricultural community was formed by Saxons, who carried out subsistence farming activities [28].

In the past, the community wood-pastures were highly valued by the Saxons, especially for animal husbandry. The grazing pressure was low as cattle and pig herds were small. Oak trees (*Q. robur* and *Q. petraea*) were selectively tended to produce more acorns, which were needed for rearing pigs. At the same time, timber and other products extracted from the pasture played a less important role than they do in the present [29].

The Transylvanian wood-pastures were most significantly impacted by the demographic changes that occurred during and after the fall of the communist system in the 1990s, as many of the Saxons left Romania [30,31]. In contrast to the strong connections between Saxon communities and wood-pastures, nowadays, these are primarily perceived as state property, having little relevance to the local population. Exceptions are the Roma-

nian farmers who use them for animal husbandry or the remaining or returned Saxons to Transylvania [29].

The wood-pastures host—as distinctive elements—ancient trees, which have multiple ecological, economic, and social functions (from habitat for countless species of wildlife to a food source for animals and local people, along with stunning aesthetics) [26,32]. However, due to changes in land use, the coverage area of these wood-pastures is shrinking, with the number of old trees gradually decreasing and not being replaced [26]. Although according to the CAP, through the Standards for Good Agricultural and Environmental Conditions (GAEC), trees on pastures must be preserved because of their importance in agrarian landscapes, they are not fully protected by current legislation, as they can be felled when they are considered to be damaged by various natural phenomena (storms, landslides, etc.), with the resulting dead wood being removed [33]. Moreover, the regeneration of trees from pastures is not mentioned in existing European legislation, with farmers receiving no financial incentives to replace them after they die and dry out due to natural causes [34].

Replanting trees on pastures has been carried out in several cases, such as Viscri and Cobor in Brașov County or Hodoșca in Mureș County [35–37]. However, a step forward has been taken in Romania by including these wood-pastures in the permanent grassland category. Under current legislation, other species used for grazing, such as shrubs and/or trees, are included in this category, provided that grass and other cultivated or spontaneous herbaceous forage plants remain predominant. Only those where the share of forest vegetation is less than 40% of the total area are considered to be wood-pastures [38,39].

Encouragingly, Law No. 97/2023 on the protection of remarkable trees was recently adopted due to Romania's potential in this respect but also because no clear record of their distribution and status existed, although there are several regional and local initiatives for creating an inventory [40].

The establishment of small farmers' associations is seen as a step forward, as pastures are leased collectively from the municipality to the association's members to use for their livestock. This ensures equitable access to natural resources for all farmers in a settlement (the concession/lease of pastures is proportional to the livestock on the holding under the legal provisions for the available grassland for a period between 7 and 10 years). The remaining unallocated grassland areas are assigned to animal breeders, authorized experts, or SMEs, with animals registered in the National Register of Holdings (RNE) [41].

Moreover, farmers can apply for subsidies from the Agency for Payments and Interventions in Agriculture (APIA) through Measure 10: Agri-environment and climate. Package 1.—High Nature Value (HNV)/High Nature Value (HNV) grassland/formerly M 214, which can thus be used for the benefit of members to maintain community-managed pastures [29,42,43]. These associations could become future communal institutions, replacing town hall governance and shaping new rules and norms tailored to socio-economic and political realities [29].

At the same time, High Nature Value (HNV) grasslands are located in Transylvanian rural areas, where highly varied biodiversity and traditional farming practices are key factors in nature conservation [44,45]. Thus, extensive grazing and traditional mowing are essential for maintaining Transylvanian grasslands' high natural value (used as pastures or hayfields).

Furthermore, High Nature Value (HNV) grasslands deserve to be supported both for conserving the natural diversity they maintain and for their economic and agricultural productivity, which provides the livelihood of farming communities [46].

In other words, traditional farming systems cannot provide the farmers with financial security or comfort for their families' livelihoods without support in terms of diversification activities in rural areas while also maintaining the balance between economic development and sustainable use of natural resources [47]. This is a significant economic, social, and environmental challenge [46].

Also, traditional farming practices maintained in Transylvanian rural areas provide a range of public benefits (goods and services), including valuable cultural landscapes,

high-quality food and water, flood control, carbon storage, recreational opportunities, etc. [48].

The recreational opportunities developed at such sites fall under the umbrella of sustainable tourism (particularly ecotourism), thus providing additional income to members of local communities. Tourists' primary motivators are getting in touch with nature and local nature-based traditions, carrying out recreational and educational activities, and having as little negative impact on the environment as possible [49]. The Transylvanian Hills region, known for its fortified churches (UNESCO sites), traditional Saxon houses transformed into guesthouses, hiking trails and cycling routes, is the newest Romanian ecotourism destination certified by the Ministry of Entrepreneurship and Tourism in 2022 [50].

It includes a series of protected areas part of the European Natura 2000 network and several overlapping wood-pastures where tourists can carry out nature-based recreational activities (idem).

In this context, the study aimed to analyze the degree of capitalization of the wood-pasture from Viscri directly related to the tourists' awareness of its importance for the local community and the environment.

Consequently, the research question was: Is the wood-pasture from Viscri used as a tourist resource? The study's objectives were: O1—Analyzing the specificities of the wood-pasture from Viscri using observation sheets and GIS techniques; O2—Identifying the tourists' level of awareness of the importance of the wood-pasture from Viscri.

### 1.3. Wood-Pastures—The Literature Review

Wood-pastures have been addressed by various studies that have considered their critical role in supporting extensive livestock farming and the threats they face [2,34,51,52]. They have also been studied in terms of their geographical distribution, socio-ecological values, ecosystem services provided, management practices, respectively, and their management in the context of the EU Common Agricultural Policies [1,7,22,53–55].

At the same time, in other studies, wood-pastures are considered an archetypal manifestation of HNV farmland in Europe [1,23,56]; their high biodiversity has been studied in Spanish dehesa [57,58], Portuguese montados [59–61] and those of the Czech Republic [62], Sweden [63], etc.

In Hungary, wood-pastures have been studied through the lens of conserving the habitat they provide for dung beetles, which has a role in the nutrient cycle. They are seen as part of agricultural systems, ensuring a socio-ecological framework for sustainable agriculture with high biodiversity [9].

In the Portuguese montados, butterfly communities have been studied in terms of the importance of the traditional management methods implemented here [64]. The largest community of spiders living directly on the ground has been identified in the Spanish dehesa in comparison to other habitat types (Mediterranean forests, pine plantations, etc.) [65].

Other works have studied either the relationships between vegetation structure and bird communities nesting in the dehesa and the influence of traditional management practices on them [66,67] or the role of birds as indicators of the high natural value of the Portuguese montado system [68].

The theme of correlating the floristic diversity of Mediterranean grasslands to traditional extensive grazing has also been addressed in articles. Thus, changes in land use, especially the gradual abandonment or intensification of grazing, are important factors leading to significant changes in the floristic composition of grasslands [69].

The literature also debates the maintenance and protection of wood-pastures, with the aim of preserving the biodiversity of these landscapes [2,7,14,70]. However, the insufficient regeneration of trees on such pastures has a negative impact on the biodiversity and ecosystem services they provide [23].

Recently, in Romania, wood-pastures have become an important research topic. The first study perspective was their role as traditional grazing systems, then as providers of

ecosystem services. Last but not least, the studies addressed the structure, condition, and threats they are subject to in the southern Transylvanian region due to land use changes, the felling of old trees, and the lack of viable strategies for regeneration [5,71,72].

The biodiversity of Transylvanian wood-pastures has been emphasized in various articles [72,73], underlying the fact that they support a rich and unique bird community that is more diverse than that of open woodlands and grasslands [34,74]. At the same time, wood-pastures have also been considered suitable habitats for amphibians such as yellow-bellied toads (*Bombina variegata*) due to their role as temporary pools (or temporary wetlands) generated by buffalo and cattle grazing [34,75,76]. They are also home to protected insects such as the stag beetle (*Lucanus cervus*) or the great capricorn beetle (*Cerambyx cerdo*) [34].

Another study addresses the issue of shrinking wood-pastures due to the abandonment of traditional agriculture and management and policy changes in the Apuseni Mountains [8]. Low-intensity farming practices in Transylvania have also been mentioned in recent research debating the issue of habitat conservation for some butterfly species, as their numbers have significantly reduced in other regions of Europe. The study shows that CAP financial support is insufficient for HNV grasslands in Romania to prevent intensive farming practices or abandonment, fostering better collaboration between local authorities, researchers, local communities, and NGOs [77]. These partnerships aim to raise awareness and encourage the participation of local community members in wildlife conservation management. One study mentions a clear urgency to develop and implement environmentally and socially feasible management plans, which, unfortunately, are currently lacking [77].

Other studies aimed to identify research gaps in biodiversity hotspots in Romanian grasslands and improve knowledge exchange between practitioners, researchers, policymakers and other stakeholders [4].

Related to leisure, researchers focused on interviewees' preferences for cultural ecosystem services. The results showed a significant preference for agroforest landscapes from rural regions as opposed to forest or agricultural landscapes, which scored lower in this respect, provided that the former were located close to urban areas or roads [78].

At the same time, according to the results of another study, experts believe that the concept of cultural ecosystem services (CES) is still far from being implemented in policies that address agricultural landscapes. The study mentions that European scientific research targeting CES in areas with HNV farmland, which also hold high cultural values, is lacking in developing regions of the continent and is only carried out in economically developed ones [79]. The study also notes that experts have highlighted potential problems with the applicability of the CES components, such as ecotourism, aesthetic values, cultural heritage and sense of place. For example, in Romania, Spain, Hungary, and the Czech Republic, CESs related to traditional cultural landscapes are currently not recognized in standard policies or among policymakers or stakeholders [79].

Thus, one can argue that the Saxon villages of Southern Transylvania have been included in various studies in terms of the richness of their ecological and cultural heritage (e.g., fortified churches, Saxon rural architecture) [80], local biodiversity of farmland, wildflower meadows and high nature value pastures [81–84], their conservation, raising awareness of farming communities towards environmental protection and developing agritourism and ecotourism in the area [85,86].

As far as wood-pastures are strictly concerned, both in Romania and in the rest of Eastern Europe, they have not yet been addressed in scientific studies from the perspective of their ecotourism development, despite the growing interest in this research topic at the European level and beyond.

In view of the above, the originality of the present study lies precisely in filling a research gap by focusing on this still insufficiently explored topic—wood-pastures as tourist resources—in this case, in the village of Viscri (Transylvania, Romania).

## 2. Methodology

The primary study methods used were GIS and remote sensing techniques, direct field observations (filling in observation sheets) and survey methods (semi-structured interviews and questionnaires). The first methodological stage involved employing GIS and remote sensing techniques to achieve the cartographic base, respectively, by processing satellite images and collecting statistical data. Subsequently, the database was built to identify and analyze the wood-pasture on the field in Viscri village. In this regard, the current situation of the wood-pasture, respectively, for 2022, was analyzed based on satellite images Landsat 8–9, OLI/TIRS C2 L1, with a resolution of 30 m [87]. The final cartographic material was processed using the ArcGIS Pro 2.8 Free Student License application [88]. The results obtained from geospatial analysis and statistical processing were correlated both with the observation sheets on the field and the information obtained from semi-structured interviews conducted with three farmers, members of the Agricultural Association "Agro-Eco Viscri-Weisskirch", for their validation.

The second methodological stage consisted of field campaigns carried out in Viscri to validate the information collected in the previous step and to collect new data needed in the study. They occurred in 2 periods: 1–5 May 2022 and 5–7 July 2022.

In the first field campaign, direct field observations were implemented, which aimed to fill in observation sheets on land use, focusing on Viscri's wood-pasture and its ecological status. Other information on the use and management of the wood-pasture was obtained from the semi-structured interviews with farmers in Viscri. The questions centered on the following issues: pasture management; pasture-raised animal species; type of grazing (intensive/extensive); current activities carried out for pasture maintenance; subsidies received from APIA for pasture maintenance and the related constraints; how local biodiversity is preserved/enhanced; if plantings are done to regenerate the trees on pasture; the benefits the wood-pasture brings to the local community and the environment; their opinion on the inclusion of the wood-pasture in the ecotourism circuit; their view on King Charles III's role in the protection and promotion of Viscri's wood-pasture; whether this promotion has benefited the local community; and the threats the wood-pasture is facing.

In the second field campaign, which took place during the peak tourist season, 168 face-to-face questionnaires were administered to visitors and tourists in Viscri. The aim was to assess their awareness of the importance of developing nature-based tourism/ecotourism for the wood-pasture in Viscri.

For the study's relevance, 300 more questionnaires were filled out online from 20 May to 24 July 2023, by people who visited Viscri. The questionnaire was designed in Google Forms and distributed on public social networks such as Facebook and Instagram, then to specific interest groups dedicated to the practice of responsible tourism in Romania (e.g., Facebook groups—What kind of tourist are you, Visit Brasov Romania, Nature education and travel with children, Kinder Trips, Vacante do it yourself). For the questionnaire distribution, the snowball method was chosen [89] to identify as many respondents who had visited Viscri as possible. This method was used after a convenience sample was identified on the field. The selection criterion for validated respondents was having previously visited Viscri, and the snowball method was based on the idea that visitors are geographically dispersed [90].

The introductory part of the questionnaire presented the aim of the research, followed by instructions that it should be passed on to other people who had visited Viscri. In accordance with EU academic research ethics, respondents were informed that by filling out the questionnaire, they had given their consent to be part of the sample analyzed and that their data would be coded, thus preserving their anonymity (Ethics Committee of the University of Bucharest, document no. 1/3 January 2024).

A total of 460 responses (obtained both from face-to-face and online questionnaires) were analyzed using Microsoft Excel Version 2310 and SPSS Statistics 23. The sample gathered 14% of the number of tourists who visited Viscri in 2019, i.e., 3326 people, according to official statistics [91]. We chose 2019 as the reference year, as the data were not biased by

the COVID-19 pandemic, which had a tremendous impact on tourist flows, travel behavior and tourism functionalities [92].

Both the novelty of the research topic in Romania, namely the less extensive research literature related to nature-based tourism on wood-pastures, and the fact that the survey was primarily conducted online explain the structure of the final sample, with a large majority of university graduates (83.1%) and mainly inhabitants of urban centers (92.75%), as Romanian rural inhabitants very rarely choose another domestic rural region as a destination (Table 1).

**Table 1.** Demographic characteristics of the sample.

| Age | 18–20 years | 21–40 years | 41–60 years | >61 years |
|---|---|---|---|---|
| | 13.10% | 53.40% | 30.10% | 3.40% |
| Sex | F | M | I do not want to declare | |
| | 68.70% | 30.90% | 0.40% | |
| Education | Primary studies | High-school studies | Higher education | |
| | 0.20% | 16.70% | 83.10% | |
| Residential status | Urban | Rural | | |
| | 92.75% | 7.25% | | |

Source: Computed by authors.

The questionnaire included 23 questions covering several general and specific issues (Supplementary Materials), with respondents having the opportunity to offer other options that they considered relevant, which allowed us to integrate their perceptions into the statistical analysis better and to create a network analysis. Respondents also provided information on various demographic variables (gender, age group, education, and place of residence) (Table 1).

Social network analysis was used, in addition to statistical frequency analysis, to identify the centrality of the benefits that grassland (pastures and hay meadows) brings to the local community and the wider environment and to establish the links between the benefits of grassland to local communities, the wider environment and the reasons for visiting/visiting grassland. This method is increasingly used for natural resource management and governance of protected areas [93] and for the identification of patterns in stakeholder collaboration in different environmental impact assessment procedures [94,95] or even in the implementation of nature-based solutions [96]. To create network matrices, we used the answers received to 3 questions (Q18, Q19, Q23—see questionnaire Supplementary Materials), thus creating networks in which the nodes were benefits for the community, benefits for the environment and reasons for return visits. The link between them was validated if respondents indicated all of them in the same answer. UCINET software Version 6.631 [97] was used for the analysis, and Netdraw 2.161 [98] was used for the graphical representation of the networks.

*The Study Area*

The village of Viscri, or Weißkirch (German name), is part of the commune of Bunești, located in the southeastern part of Transylvania, Romania (Figure 1). The locality is a small one, with a population of 450 inhabitants in 2021, but with a large number of tourists, i.e., 4465 in 2021 [91]. One cited reason for this is that it is one of Transylvania's most beautiful and authentic Saxon villages, present on the UNESCO World Heritage list [99].

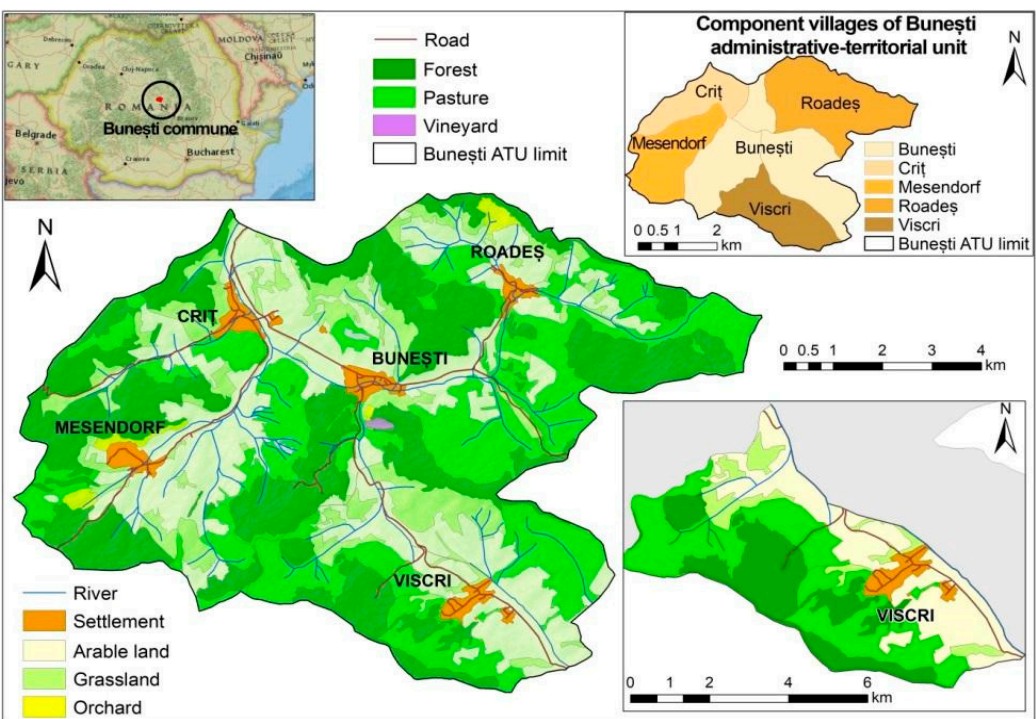

**Figure 1.** Viscri village, Bunești commune, Transylvania (Romania). Source: Aerial Images, 30 m resolution, Google Earth Pro.

Its landscape is hilly, with the village's countryside bordered by arable land, meadows with wildflowers, pastures, wood-pastures and forests, which have a very high level of biodiversity as a result of using traditional agricultural practices over time, which have maintained the farmland and meadows at a high natural value [100–102]. At the same time, two Natura 2000 protected areas overlap the territory of the locality, namely the Sighișoara—Târnava Mare Site of Community Importance (SCI) ROSCI0227 and the Podișul Hârtibaciului Special Protection Area (SPA) ROSPA0099 [103].

The main occupation of the locals is subsistence agriculture—animal husbandry—and a series of small creative industries rooted in the local realities (e.g., homemade cheese and jams, wool weaving, felting, masonry, ironwork, etc.). Tourism entrepreneurship has recently become an alternative, as in recent years, several guesthouses and small slow-food restaurants have appeared in old Saxon farmhouses, where tourists are welcomed with local, traditional products [102,104,105].

Although on the territory of other Transylvanian localities there are more extensive wood-pastures with ancient trees, we have chosen as a case study the wood-pasture in the village of Viscri, as it benefits from more assets compared to the other mentioned territories, namely tourism promotion, the use of sustainable agricultural practices and community management for the benefit of small farmers.

## 3. Results

### 3.1. The Wood-Pasture in Viscri—Brief Analysis Based on Observation Sheets, Semi-Structured Interviews and GIS Techniques

The wood-pasture in Viscri is part of the pastures existing on the territory of Bunești commune, to which the Viscri village belongs, according to the land use analysis for 2022 carried out by GIS techniques (Figure 2).

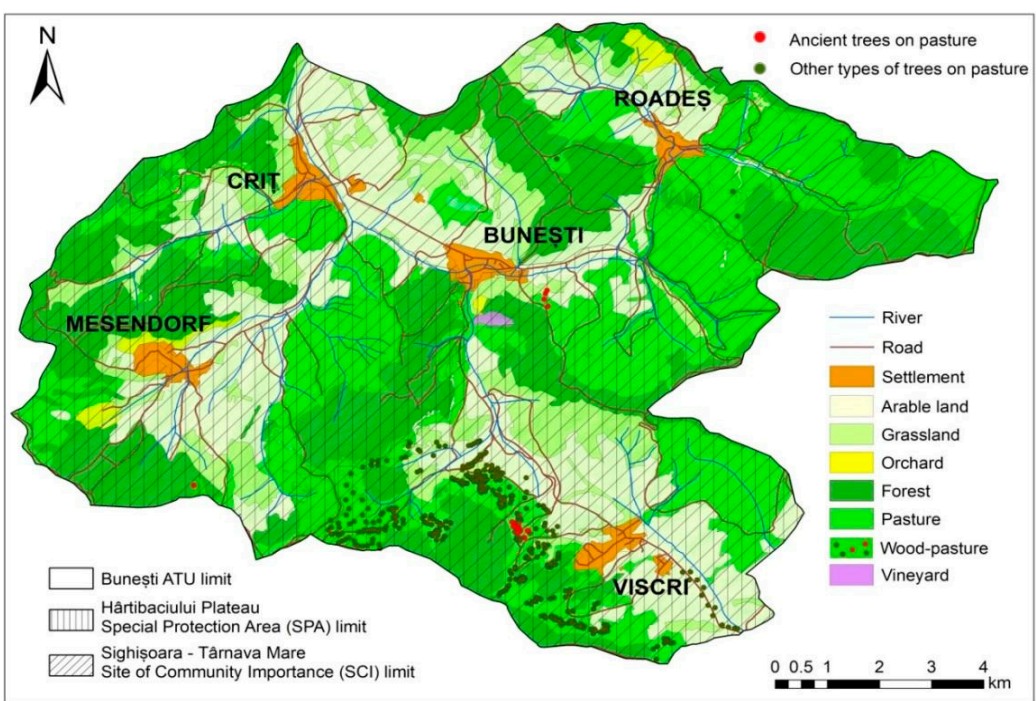

**Figure 2.** Land use in Bunești commune in 2022. Source: Aerial Images, 30 m resolution, Google Earth Pro.

It is located in the western part of the village and occupies an area of 706.13 ha (34.46%), of which 140.40 ha (6.85%) is a pasture with ancient trees. Access to the wood-pasture is made via an unmarked forest road, which intersects with the county road (DJ 104 L) connecting the villages of Viscri and Bunești (Figure 3).

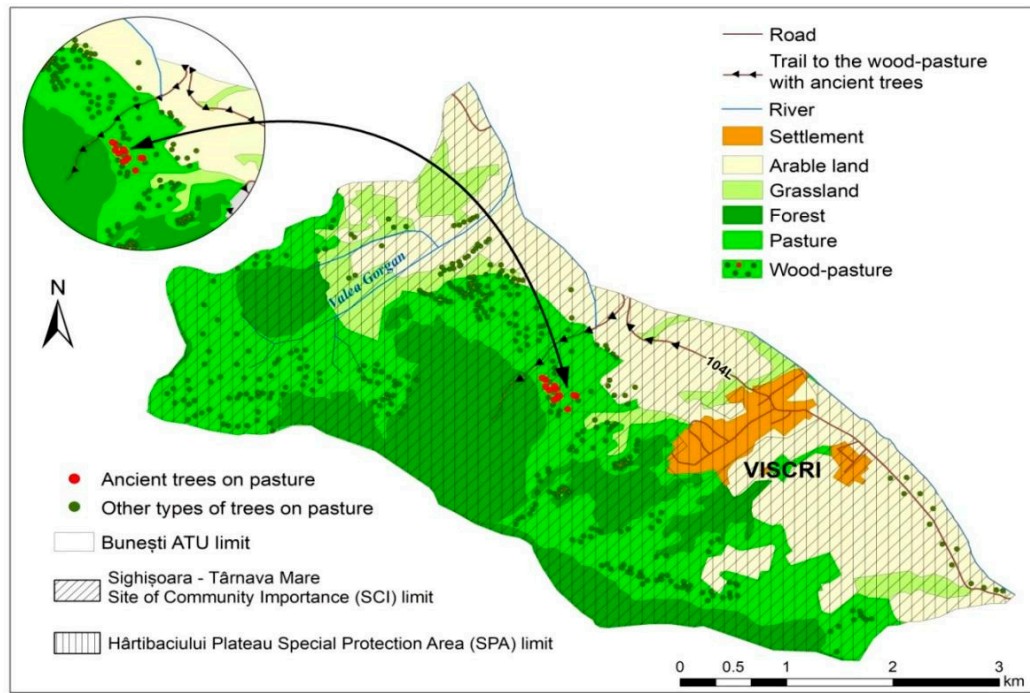

**Figure 3.** Location of the wood-pasture of Viscri. Source: Aerial Images, 30 m resolution, Google Earth Pro.

The conclusion drawn after analyzing the observation sheets and the land use map was that the wood-pasture with ancient trees was formed over time as a result of low-intensity human activities and extensive grazing practiced since the Middle Ages by the Saxons who colonized the region. It hosts species of trees such as pedunculate oak (*Quercus robur*), sessile oak (*Quercus petraea*), beech (*Fagus sylvatica*), hornbeam (*Carpinus betulus*) and lime (*Tilia europea*), with an average circumference of 429.64 cm, scattered on the pasture among other younger trees and shrubs (Figure 4).

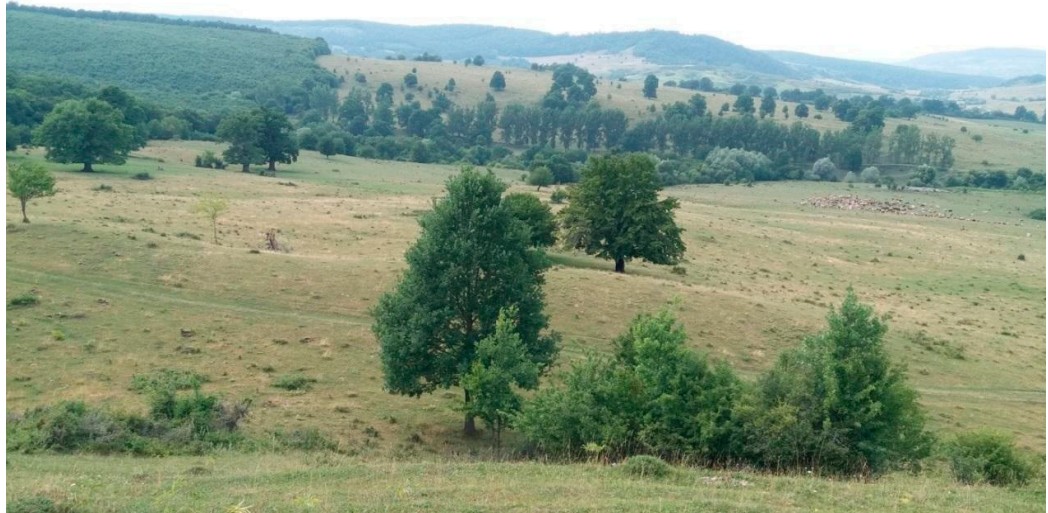

**Figure 4.** The landscape of the wood-pasture with ancient trees from Viscri. Photo: Vijulie I. (2022).

The floristic (e.g., grasses, legumes, wildflowers—*Ajuga genevensis*, *Rhianthus minor*, *Dictamnus albus*, *Dianthus carthusianorum*, *Salvia nutans*, *Orchis morio*) and faunal diversity of the pasture (e.g., insects, butterflies, grasshoppers, amphibians, reptiles, birds, mammals, etc.) are also maintained as a result of the use of traditional environmentally friendly agricultural practices.

It is precisely this diversity that led to the inclusion of the wood-pasture among the High Nature Value (HNV) landscapes overlapping with two Natura 2000 sites (Sighișoara—Târnava Mare Site of Community Importance and Hârtibaciului Plateau Special Protection Area) (Figure 3).

Further relevant information was retrieved from the interviews with the three members of the farmers' association. According to them, this pasture is the only one managed communally in the southeastern part of Transylvania, following an old Saxon model. It is administered through the agricultural association "Agro-Eco Viscri-Weisskirch", founded in 2015. This agricultural association aimed to support farmers in terms of animal husbandry, facilitating access to the pasture for all livestock farmers according to the total number of animals owned within the association. In this way, farmers with few animals would not be excluded. Equitable access to resources for all villagers has mitigated farmers' need to give up their primary occupation—animal husbandry. At the same time, the association's lease of the pasture has also been beneficial in continuing traditional farming practices, particularly extensive grazing, which has prevented either the possible abandonment of the pasture or its excessive use, which would have negatively impacted the local biodiversity.

> "*We are careful that both man and nature win. . . to us, intensifying agricultural practices or abandoning them is not a priority; we always choose to keep a balance, to graze extensively*" (excerpt from the interview with a 70-year-old male farmer from Viscri).

Pasture management in terms of use and maintenance involves a series of rules to be followed by all members of the association, plus a series of regular meetings where pasture management is discussed and where "nature-based solutions", according to the EU Strategy on Promoting Biodiversity, are supported by the whole community. Each livestock

owner participates in pasture maintenance a given number of days a year, depending on the number of cattle or sheep/goats he owns.

Current practices/activities farmers use for pasture maintenance are traditional: the soil is not leveled, fertilization is done with manure, brambles are removed, and thorny shrubs are selectively maintained (e.g., hawthorn, rosehip, blackthorn). On the one hand, they work as natural facilitators for future trees that naturally settle on the pasture and, on the other, as places for birds' nests. In autumn or spring, there are campaigns to plant saplings for tree regeneration and maintenance, which are protected from cattle with natural "fortifications of thorns".

> *"The mechanical mower is very useful for selectively removing bushes and scrubs from the pasture in the spring, as with the help of toothed rollers less pressure is exerted on the soil"* (excerpt from the interview with a 65-year-old male farmer from Viscri).

> *"The woody vegetation spread on the pasture increases its biodiversity, so we need to take care of it… For us, for example, shrubs are very important because their thorns keep away cattle/cows, so we can naturally regenerate at least some of the trees on the pasture. Our ancient oaks provide the highest quality acorns; they have lasted for hundreds of years here, so they have outstanding genetics, adapted to our region"* (excerpt from the interview with a 60-year-old male farmer from Viscri).

As mentioned above, this ancient management system, based on the Saxon model, is also supported by APIA compensation payments. However, according to the farmers, these are insufficient.

> *"Subsidies are not enough; we nowadays get the same amount as in 2016; we receive 142 EUR /ha/year for high nature value grasslands, and 100–121 EUR /ha/year for traditional farming practices, depending on how we carry out the maintenance work, manually or with light machinery… we are unhappy because now everything has become more expensive"* (excerpt from an interview with 58-year-old male farmer from Viscri).

The subsidies received from APIA for pasture maintenance through Measure 10. Agri-environment and climate. Package 1.—High Nature Value Grasslands (HNV); Package 2.—traditional agricultural practices (applied only in combination with Package 1) are welcomed by farmers. Even if they involve several constraints (e.g., no use of pesticides and chemical fertilizers, no plowing or disking of the pastures, no surface seeding or overseeding, etc.), the farmers' association follows these prerogatives because they perceive the subsidies as a means for sustainable local economic development and continuous transfer of traditional agricultural practices.

For the same pasture (permanent grassland according to APIA classification), the Association "Agro-Eco Viscri-Weisskirch" also receives annual subsidy/compensation payments from APIA through Measure 11. Organic farming (financial support for maintaining organic farming practices and methods), amounting to EUR 39/ha/year, as long as they also follow Measure 10.

The farmers also mentioned the presence of the Agricultural Development and Environmental Protection in Transylvania Foundation (ADEPT), which runs several projects in Transylvania, including in Viscri, and tries to identify different opportunities to reward local community members for sustainable landscape management, respectively, to set additional agri-environmental measures to provide financial incentives to farmers for pasture conservation.

In this regard, this foundation and Stefan cel Mare University of Suceava implemented the European project EFFECT between 2019 and 2023. The project aimed to improve the methods of providing financial incentives for biodiversity conservation through agri-environmental measures. Thus, it sought to implement an innovative results-based payment scheme, which proposed that subsidies to farmers should be conditioned by the preservation of plant species identified on the pasture.

Around the same time (2020–2023), the farmers of Viscri also participated in the Life TransilvaCOOPERATION pilot project, funded by the EU LIFE Programme and im-

plemented by the same ADEPT Foundation, which aimed to conserve biodiversity-rich grasslands within Natura 2000 sites through traditional farming practices, generating additional income for farmers through better quality products.

The ADEPT Foundation is also currently implementing the LIFE project "Metamorphosis" (2022–2029), which has as a case study the Site of Community Importance ROSCI0227 Sighisoara—Târnava Mare that overlaps with the study area and seeks to restore habitats of butterfly species (*Colias myrmidone*), found on meadows, pastures, forests, etc. This new project intends, on the one hand, to develop best practices for the conservation of butterfly species by creating agreements with farmers' associations managing the land and, on the other hand, to collaborate with local communities to develop management plans for the landscapes they sustainably manage by introducing reward schemes for farmers.

> "*The ADEPT Foundation is supporting us and looking for solutions so that we can receive higher subsidies from the EU as we strive to conserve our landscapes inherited from our ancestors...*" (excerpt from an interview with farmer, male, 65 years old, Viscri).

When it comes to the farmers' knowledge of good management practices, some of them have been inherited from their elders (e.g., traditional, extensive techniques that do not harm flora or fauna). In contrast, others have been acquired as a result of the farmers' association's collaboration with ADEPT Foundation specialists or due to their openness to collaborate with higher education institutions with expertise in the field (e.g., Faculty of Forestry, Stefan cel Mare University of Suceava; Faculty of Environmental Science and Engineering, Babeș-Bolyai University of Cluj-Napoca).

At the same time, the farmers consider that King Charles III plays a crucial role in protecting and promoting the wood-pasture in Viscri.

> "*King Charles III has promoted our natural and cultural values a lot, and we are forever grateful for this... he loves our pastures, the wildflowers, and the old trees.... he brought only benefits to our community after he bought a house in our village and got involved in nature conservation and traditional local farming*" (excerpt from an interview with a farmer, male, 70 years old, Viscri).

The threats this ancestral landscape faces come either from overgrazing, the abandonment of pastures, or the tendency to develop more intensive agriculture practices with large monocultures (so-called "green deserts"), which could endanger the regeneration of rare plants, leading to their disappearance. According to our results, farmers were aware of these threats. They were informed on these topics through various workshops held in Viscri, including one held by John Akeroyd, King Charles III's botanist, and his team.

*3.2. Identifying Tourists' Level of Awareness about the Importance of Viscri's Wood-Pasture Ecotourism Development*

3.2.1. Becoming a Destination—Wood-Pastures as a Basis for Nature-Based Tourism Development

The most cited reasons for choosing to visit the Viscri area were curiosity, the scenery, relaxation, sightseeing, outdoor walks, nature's and region's beauty and uniqueness, touring the fortified churches, the bike trails, the famous beauty of the locals' houses, the scenery and history of the place and the fortified evangelical church, but also following the popularization of the area in the media and seeing what King Charles III's favorite Romanian destination looked like.

A significant result is that most respondents (86.4%) visited the area because of its cultural attractions. This is of particular importance for this study as it proves that choosing to visit Viscri's site for its wood-pasture takes second place, with respondents selecting the area primarily for nature relaxation (37.1%—nature activities such as hiking its hilly terrain, bird watching, picnicking, visits to local stables) or local gastronomy (29.7%), and placing as less important activities such as cycling (7.10%), event tourism (6.30%—local celebrations such as *Festivalul florilor de camp—Wildflower festival, Săptămâna Haferland/Săptămâna Țării Ovăzului—Haferland Week/Oat Country Week, Cununa florilor—Wreath of Flowers*, the village

gastronomy festival, cycling competition, etc.), horse riding (4.70%) and other options (2%—such as exploring the forest and pastures of the area accompanied by truffle pickers and their trained dogs, visiting the brick factory, seeing the beautifully renovated Saxon houses, walking through the village, rural atmosphere, seeing old houses, discovering local handicraft products, etc.).

As Figure 5 shows, respondents rated the attractiveness of the wood-pasture in Viscri, i.e., the hills (as the locals call them), which are rich in biodiversity rarely found elsewhere in Europe (e.g., wildflowers, scattered trees, insects, butterflies, animals), as extremely attractive (56%); overall, more than 99% answered, placing them in a position above average when it comes to their appeal.

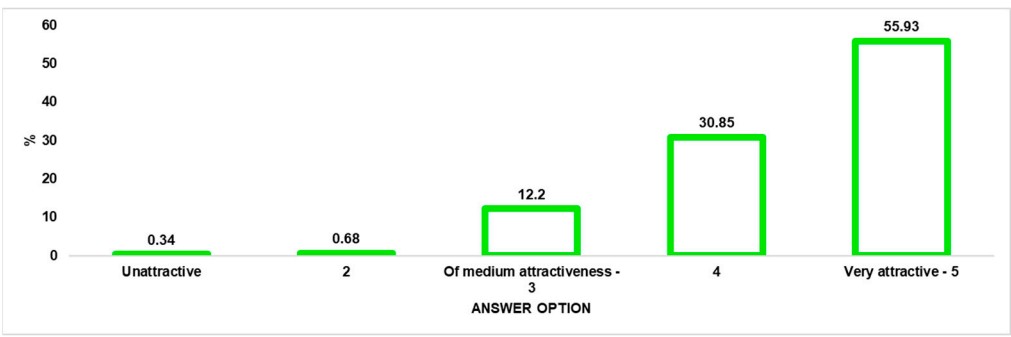

**Figure 5.** Viscri's wood-pasture attractiveness.

Surprisingly, although they had visited the area, 45% of respondents said they did not know about the invaluable wood-pasture and wildflower meadows. Of those who declared to have information on this topic, 26.9% said they learned about these habitats from mass media, 18.6% from their hosts or other locals in Viscri, 12.9% from promotional materials on social media, 12.2% from friends and 3.1% from close relatives or family.

As this is a well-established fact in the Romanian mass media's [106–110] coverage of Viscri, respondents were asked to rate how much King Charles III's connection to the Viscri name has contributed to promoting the area as a tourism destination. Thus, 44% of the respondents rated King Charles III's contribution as very important, and 32% rated it as very important. At the other end of the scale, only 2% of the respondents considered that he had not played an essential role in the area's tourist development.

It is worth noting that respondents considered that the mere fact that King Charles III is a world-famous personality attracted interest in the area where he chose to purchase a holiday home. Other cited reasons for his contribution to increasing the area's fame are the fact that he loves nature and the area's native biodiversity; he appreciates the countryside and traditional agriculture; he is fond of the local architecture and specific crafts such as weaving, felting, masonry, ironwork and mowing (Figure 6).

At the same time, respondents considered that the main reasons why King Charles III contributed to the promotion of the wood-pasture in Viscri are that he supported the project "Arbori remarcabili din România—Remarkable trees in Romania" and advocated for the maintenance of wood-pastures throughout Transylvania due to their biodiversity. Some people also felt that it was not consequential that he is an environmental activist or advocate for the conservation of vulnerable plants in the face of climate change, but in a very low percentage, as seen in Figure 7.

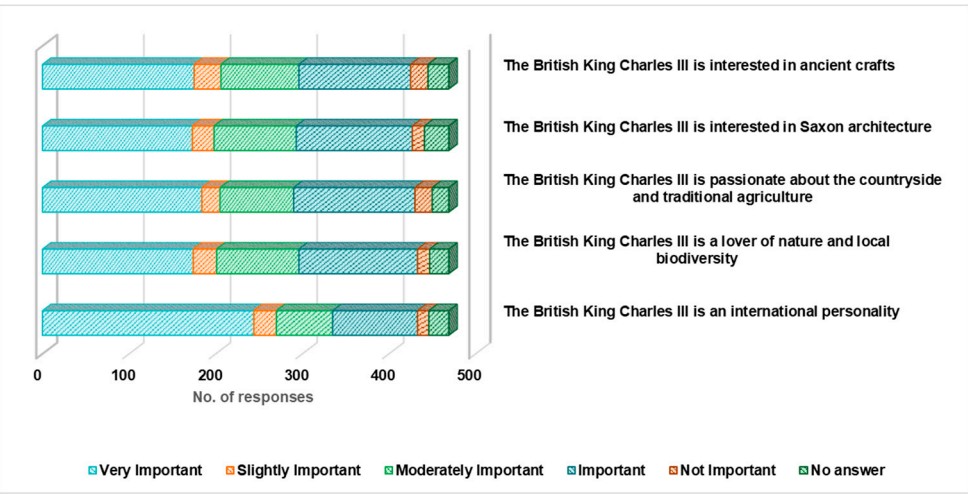

**Figure 6.** Perception of the reasons why King Charles III contributed to promoting the village of Viscri.

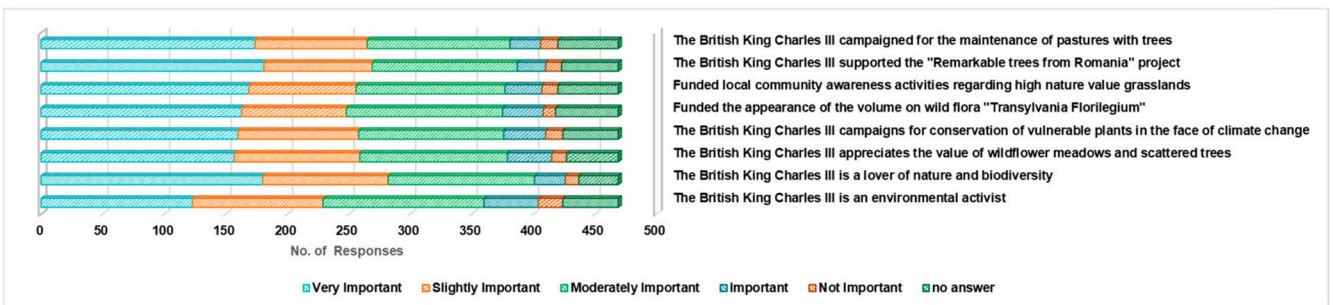

**Figure 7.** Appreciation of the importance of the reasons why King Charles III contributed to promoting the wood-pastures of Viscri.

### 3.2.2. The Road to Ecotourism

When questioned about the types of activities suggested to them by their hosts, a majority of responses indicated that no such recommendations were made. Most visitors and tourists choose relaxing activities in nature based on their preferences and needs. The most common activities they carried out during their visits to the wood-pasture were as follows: observing old trees, animals, butterflies, insects and flowers (28%); taking photographs and nature observations (24.1%); rides in traditional wagons on the pasture (13.3%); picnics with organic products—bacon, cheese, cheese, homemade bread, vegetables, fruits (13.1%); hiking in the hills with their children so that the latter ones would be in contact with nature and learn to respect and value it more (9.7%); "mushroom hunting" with specialized guides (9.1%); bike rides (9.1%); wagon rides to stables located on the pasture to take part in a shepherd's meal containing pastrami, polenta and local cheeses (6.4%); hiking in the hills both for connecting with nature and for scientific purposes (botanical, faunal) (6.4%); participating in festivals (5.8% of the respondents had experienced the "Festivalul florilor de câmp", centered on meadows full of wildflowers, birds and animals, which have disappeared in other parts of Europe, but live harmoniously on the farmland in Viscri); horse riding (5.1%); general observation of the countryside from the fortified church tower (0.2%) (Figure 8).

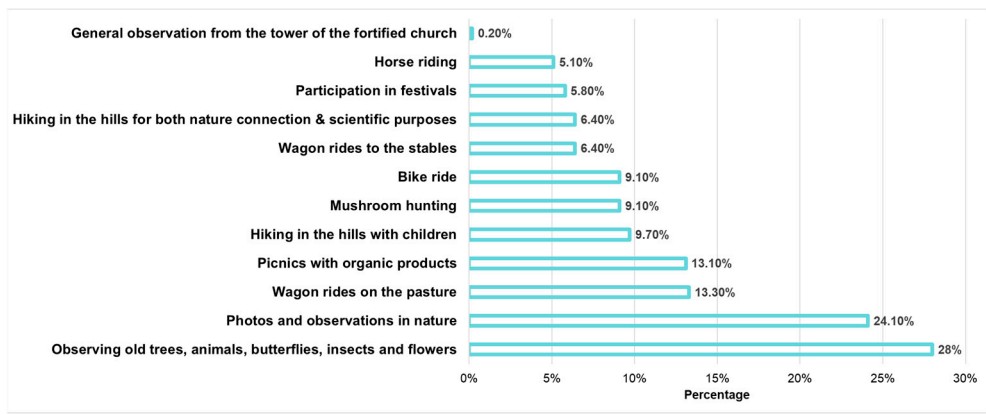

**Figure 8.** Most common recreational activities carried out during visits to the Viscri wood-pasture.

Almost 92% of respondents recognized the importance of Viscri wood-pasture's ecotourism development (Figure 9). Tourists mainly mentioned that the landscape here is very advantageous for ecotourism activities because the village of Viscri itself attracts more and more tourists who are passionate about nature or outdoor activities and who want to discover unique landscapes (hiking, horseback riding, wagon or bicycle rides on hills with pastures and meadows), and last but not least because Viscri is part of the Transylvanian Hills ecotourism destination.

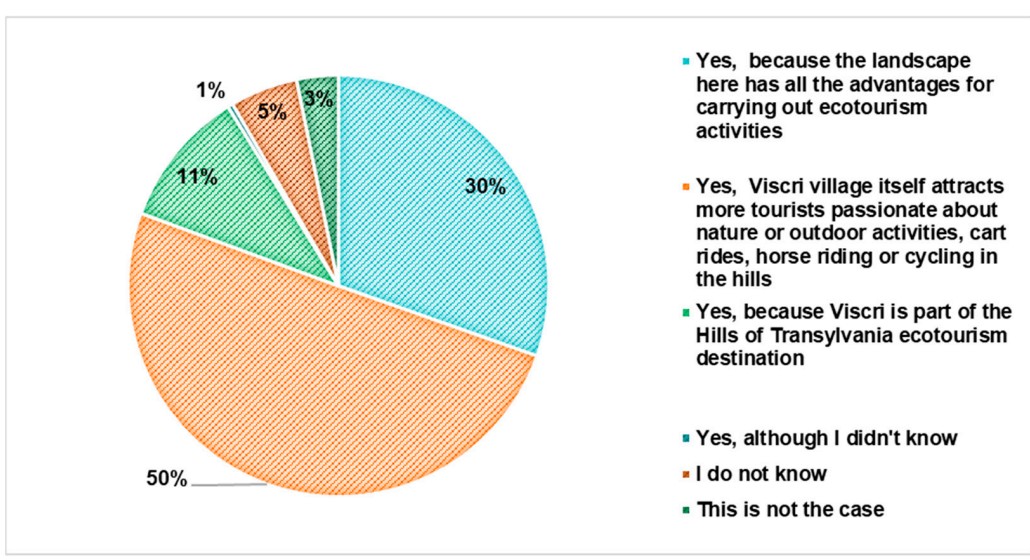

**Figure 9.** Perceived importance of the Viscri area wood-pasture' ecotourism capitalization.

3.2.3. Prospects for the Capitalization of Wood-Pastures

Respondents placed as the most important recommendations for environmentally friendly activities that could be carried out on this wood-pasture (Figure 10) the following: establishing themed trails for nature observations (e.g., brightly colored wildflowers in the meadow, old trees scattered in the meadow, grazing animals, butterflies, insects, etc.); picnics with local natural, organic products; country festivals (e.g., "La cosit—Mowing", "Festivalul florilor de camp—Wildflowers Festival"), bicycle or wagon rides.

Also, as indicated by the respondents, the area could attract tourists due to the peacefulness of the meadows, the emotions they stir, the well-being generated by these ancestral landscapes, the colors of nature, which change from one season to another (idyllic landscape), the smell, the clean, unpolluted air, the fairy-tale sunsets and the birdsongs.

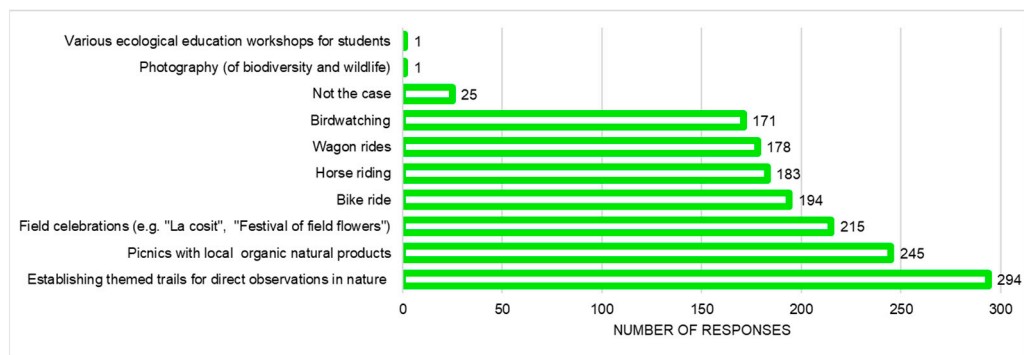

**Figure 10.** Recommendations for environmentally friendly activities to be organized on the wood-pasture.

Respondents also mentioned that some tourists might be enthusiastic about rediscovering "the pasture similar to the one they knew during their childhood" (a collective cultural memory, reminiscing about their activities on their native pastures with their parents and grandparents).

Tourists might be inclined to visit this wood-pasture for other reasons, including medicinal plants/aromatic plants, traditional cooking courses, participatory activities in the life of the village, photography (of biodiversity and wildlife), and various ecological education workshops for students.

Among the most often cited benefits that the wood-pasture brings to the Viscri local community, according to the field research, are the following: financial ones, as a result of carrying out ecotourism activities (e.g., providing transport for tourists on the pasture for various activities—horse-drawn wagon rides, bicycle rides, horse riding, rides to the stables, organizing picnics, providing local food for picnics, etc.), organic products (milk, cheese, meat, etc.), the possibility of selective collection of wild products (mushrooms, wild fruit—wild apples, pears, strawberries, rose hips, berry catchfly, etc.), fodder for animals (hay, foliage, hornbeam) and, why not, shade for the animals on the pasture.

In enumerating the environmental benefits of wood-pasture (Figure 11), the most prevalent one relates to providing a habitat for many living creatures. This translates into a shelter for wild animals and birds (burrows, nests, etc.), including dead wood (necessary for sustaining biodiversity), as well as the fact that these micro-ecosystems contribute to increasing the local biodiversity (species of butterflies, insects, birds, amphibians, reptiles and mammals), improving the local microclimate and stopping soil erosion and carbon storage.

Moreover, the level of satisfaction after a visit to the Viscri wood-pasture was predominantly high, with a combined proportion of 59.6% reporting either very high or high satisfaction. As shown in Figure 11, the benefits provided by the pasture to the community and the environment are closely related to the reasons for visiting/revisiting the area. When asked if they intend to revisit the wood-pasture, 73.6% of respondents said they would, and 53.9% said they would recommend it to their acquaintances. They justified their desire to revisit the pasture for the following reasons: for its nature (wild/field flowers, old trees, animals, birds, insects), for the fresh air, for relaxation, for the local gastronomy, for another visit to the sheepfold, for the outdoor activities (cycling, horseback riding, hiking), for the "mushroom hunt" experience in the hills, for exercising their passion for photography and nature observations and for carrying out activities aimed at educating the younger generation in the spirit of protecting nature and connecting with it. They also mentioned that nature is vital to children's development on all levels: intellectual, emotional, social, spiritual and physical, and that they would gladly return for the stories and legends local guides tell about this place (e.g., Oatland/Haferland in German), but also for scientific interest (e.g., botanical, faunal).

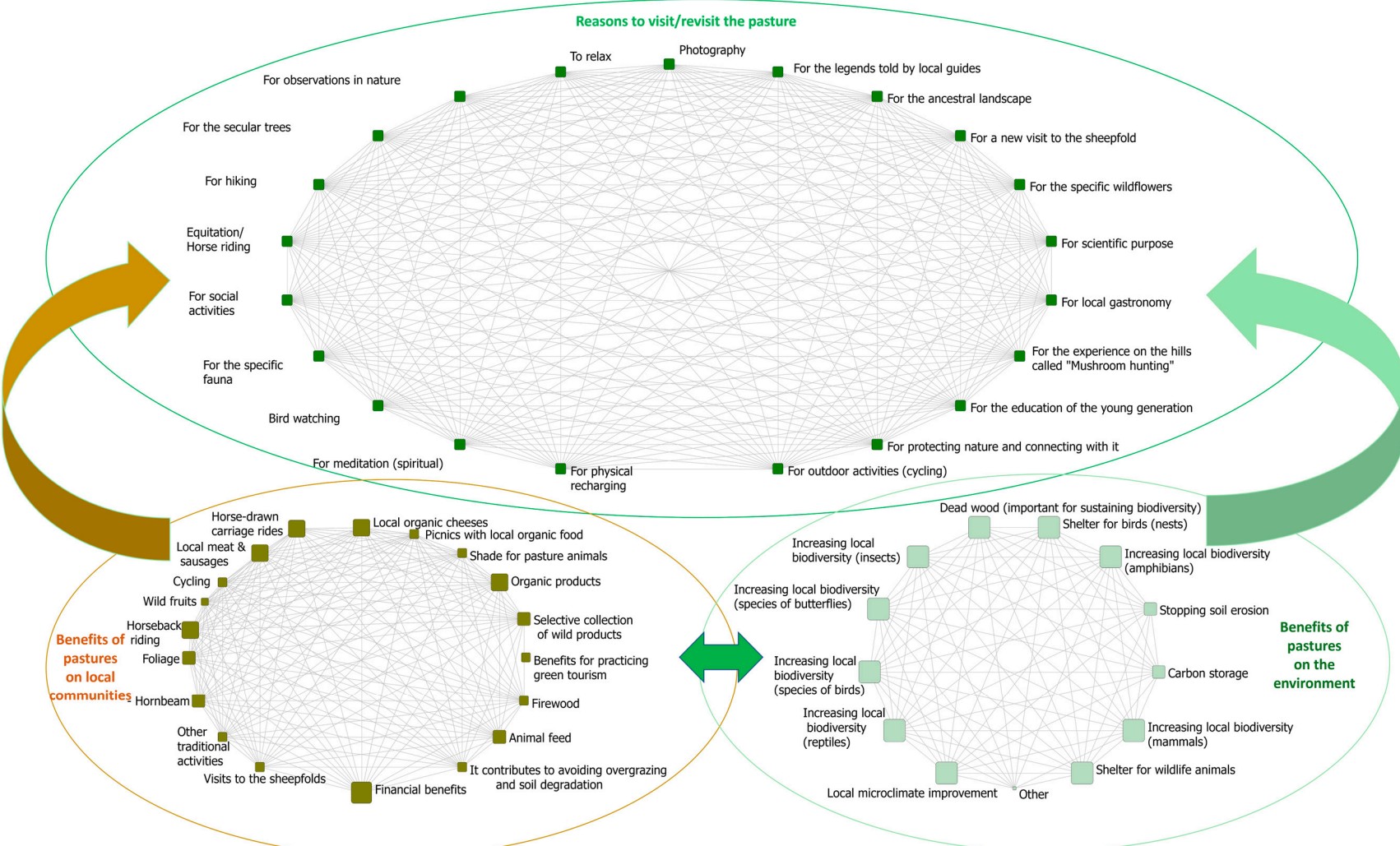

**Figure 11.** Connections between the benefits of pasture for local communities, the environment in general and reasons to visit/revisit the pasture.

## 4. Discussions

### 4.1. Viscri Wood-Pasture—A Brief Overview

As these research results have shown, the wood-pasture in Viscri, managed communally according to the Saxon model, proved to be worthy of being followed by managers of other pastures of this type from Romania, despite the abovementioned shortcomings related to the insufficient subsidies provided by the state. Their advantages come from the perspective of limited use of natural resources over time because of subsistence agricultural activities, which have reduced pressure on soil, flora and local fauna.

Different studies have also analyzed the importance of the biodiversity aspects of wood-pastures [34,72–76].

The pasture is also subject to a restoration process, as, according to the results obtained, portions without trees identified by the farmer members in the agricultural association of Viscri are significant for bringing back scattered trees in order to increase local biodiversity, an aspect that has also been studied in other European regions, where such pastures are reintroduced to conserve and increase biodiversity [2,9,14,15,23].

Meanwhile, in other European regions, the pasture management system is more efficient, especially because farmers are more aware of the importance of biodiversity conservation and, hence, willing to get involved in a more sustainable use of resources. For example, Switzerland applies the silvopastoral ecosystem-oriented exploitation and conservation model that prioritizes ecological diversity [111].

Spatial transformation in the wood-pasture landscape is determined by socioeconomic opportunities and limitations in agriculture and forest policy [112,113].

Therefore, the community use of wood-pastures is a common practice in the areas where they have been preserved, while conservation regulations were introduced quite recently in the context of sustainable development. In terms of tourist use, not all these landscapes benefited from promotion. However, tourism is a viable alternative (also from an ecological point of view), bringing significant contributions to the community, as it happens in Viscri.

The Viscri wood-pasture belongs to the category of Transylvanian pastures, hosting the highest number of flora and fauna species in Europe, and is considered unique in this respect and protected under the EU Nature Directives [114,115]. Results show that pastures like this may be subject to multiple threats (overgrazing, abandonment, and intensification of agriculture). For example, too much fertilizer or too many animals sent to graze on the pasture could make it impossible for less competitive species like orchids or other wildflowers to survive. According to different studies and research, such alarm bells have already been sounded at the European level, where such landscapes have disappeared or are on the verge of extinction [1,7].

The main way to reduce the multiple threats to wood-pastures is an increase in the financial incentives provided by the EU to farmers to maintain these ancestral landscapes, a point also mentioned and argued by other studies in Europe [2,9,77]. Last but not least, recreational activities carried out in nature on wood-pastures, are also very important as they generate auxiliary income for local community members. In addition, the preference of urban respondents for agroforestry/agricultural landscapes in rural areas has also been mentioned by other authors [78].

### 4.2. Becoming a Destination

Thus, Viscri has also become an attractive tourism destination, especially for people from urban areas, which sustains the predominantly urban and very low rural sampling for the questionnaire target group. According to the researchers who focused on the push and pull motivations, most rural tourists search for peace, solitude, and change from routine and traditional food [116,117]. Beautiful landscapes and outdoor activities are also among the most frequent motivators [118]. Tourists from rural areas are very few, as the population is aging in Romanian rural regions [119], and the average income of the elderly is relatively low (324.85 EUR /month in 2021) [120]. While some rural dwellers can afford a holiday,

they usually choose urban destinations in the country or go abroad because the rural landscape has roughly the same values/characteristics throughout Romania and is not very attractive from a tourism point of view for people living in rural areas, as tourism activities are seen as a break from routine [121–123].

When questioned on the accommodation options in the area, most respondents did not stay in Viscri, being day visitors (58.6%). Of those who did spend at least one night in Viscri, they chose a tourist guesthouse, a homestay or accommodation with acquaintances in the village, with the rest opting for a hotel, Airbnb or a tent in another locality. This prevalence of day visitors and no overnight stays can be explained twofold. Firstly, there are few accommodations available in Viscri, i.e., eight agritourism guesthouses [91], and secondly, the prices for accommodation in Viscri are usually above the budget of most Romanian tourists (100–120 euros/room/night). This aspect can also be linked to the aim of practicing sustainable tourism based on simplicity, uniqueness and authenticity, which does not encourage mass tourism.

However, the higher number of tourists registered for Viscri between 2020–2021 compared to 2019 shows that Romanian tourists opted for domestic holidays to more remote, safer areas, with accommodation in villas, thus reducing travel distances by choosing destinations closer to home areas due to the pandemic context.

Another result worth discussing is that most survey respondents who came to Viscri chose to visit mainly cultural sights (e.g., the fortified church, the house of King Charles III). On the one hand, this can be justified by the fact that they can be included in a one-day itinerary and, on the other hand, because they are more heavily promoted than the wood-pasture. Additionally, both the fortified church and the house of King Charles III, which is open to visitors, are part of the UNESCO Heritage, hence the respondents' much greater interest in visiting cultural sites in the village. However, even by simply visiting the house of King Charles III and the converted farmhouse, which houses the permanent exhibition "*The Transylvania Florilegium*" visitors can admire the flora of the region painted by botanical artists, plus the royal album, which depicts the rare and endangered plants of Transylvania. The album was commissioned and prefaced by the king himself, a passionate supporter of biodiversity conservation and environmental protection. King Charles III's project carries the important message of learning to value what we have, such as the unique flora of Transylvania.

Supporting the lower number of visitors and tourists to the wood-pasture, we bring into discussion the fact that, in this case, we are also talking about the practice of a niche type of tourism, which includes urban respondents with higher education who are passionate about observing, studying and protecting the environment.

*4.3. The Road to Ecotourism*

Although most hosts did not propose any activities to participants in this study, there were some suggestions for leisure activities, such as a wagon ride or a bike ride in the hills, visiting the sheepfold, hiking, admiring the local traditional houses, horseback riding, visiting the toasting shop, visiting the bread oven, learning a traditional craft (sewing or blacksmithing) or having a brunch with locals, sampling organic products. Among the few recommendations made by the hosts, there were, however, activities related to outdoor activities in the pasture, but not enough.

Therefore, the wood-pasture in Viscri should benefit from more tourism promotion, as it has the necessary assets to attract a greater number of nature lovers interested in relaxing outdoor activities in the countryside, a result also justified by other studies [78]. At the same time, the route to the pasture is not sufficiently marked, as there are no tourist information panels, and the locals are the ones who guide tourists to the forest road leading to the pasture (Figure 3) when the latter go hiking unaccompanied by guides or local, knowledgeable people (e.g., carters).

Although King Charles III promoted the wood-pasture both personally, in his countless annual visits or by supporting the "Arbori remarcabili din România/Remarkable trees

from Romania" project, as well as through his foundations, we believe that local authorities should also show interest in this pasture and should see it as a good opportunity for community development.

There are many wood-pastures in Transylvania that are not well managed and preserved by the local community (e.g., Rupea, Daia, Saschiz, etc.), which completely lack any kind of promotion, much less the "additional" one carried out by King Charles III. Therefore, most of them are not as sought-after as the one in Viscri.

However, there are two other wood-pastures in Transylvania whose promotion and capitalization of ecotourism are managed by local NGOs: the one in Ticușu (cycling routes, birdwatching, botanical tours, cart rides) and the one in Mercheașa (walking and cycling routes, marathons). Their programs are similar to those in Viscri, aiming at biodiversity conservation, replanting trees on pasture, pasture maintenance and promotion of ecotourism activities, even if they do not benefit from the "royal" brand.

*4.4. Prospects for Capitalizing Wood-Pastures*

As current research results have shown, the wood-pasture in Viscri provides several tangible benefits [124] for both the local community and the environment, which are interlinked (Figure 11). Although tourists and visitors are more likely to recognize the direct benefits to the community, there is a need for a greater understanding of the numerous environmental benefits that people associate with this characteristic grassland landscape, such as halting soil erosion or carbon storage. Interestingly, many of the identified benefits are also closely related to the reasons chosen by tourists for revisiting the Viscri wood-pasture.

We continue to support the idea that insufficient ecotourism valorisation of the wood-pasture could represent a disadvantage for the local community, as the income from ecotourism, together with other financial incentives, is not yet sufficient to supplement the local farmers' constraints to preserve the high-value biodiversity of the wood-pasture (e.g., its exclusive use for traditional practices), according to other studies, projects and programs carried out in the area with the aid of European funds [125–127].

*4.5. Recommendations*

We recommend that relevant authorities promote the biocultural values of the Viscri wood-pasture beyond Romania's borders in order to attract more tourists (e.g., scientists, master's students, PhD students) interested in scientific tourism and research and development, as these types of tourism activities have already attracted researchers from abroad. They can visit the pasture to study its biodiversity and exchange experiences on best practices, as the species of plants and wildflowers here, along with species of butterflies, locusts, insects, birds, reptiles, small mammals, etc., have disappeared or considerably reduced their range in other parts of Europe. It should be noted that this form of specialized tourism is in its infancy in Romania and is practiced predominantly in the Transylvanian Saxon villages, where foreign visitors have begun to study the grasslands' biodiversity. At the same time, we recommend organizing guided tours on thematic trails, which can be covered by bicycle or on foot by ordinary tourists.

*4.6. Study Limits*

The present study also had some limitations. The first limitation was that the elderly group (over 61 years old) could not be well represented in the sample because most Romanian seniors are not sufficiently involved in tourism activities due to their low income compared to the European average. A second limitation of the study was that it was impossible to obtain a balanced picture of the visitors and tourists interviewed concerning the urban-rural environments. Tourists in Romania generally come from urban areas, as their financial possibilities are better than those living in rural areas. Additionally, another limitation we mention is the potential for non-response bias, which may affect the reliability of our findings. A final limitation of the study was that official statistics only show the total number of tourists staying in hostels (or other accommodation structures), and there

are no data on the total number of tourists visiting Viscri, which is also highlighted by the study results.

## 5. Conclusions

The Viscri wood-pasture is community-managed and maintained through traditional practices according to ancient Saxon rules passed down from generation to generation. This characteristic has been made possible in modern times by the establishment of the village farmers' agricultural association, which aims to restore it and maintain its biodiversity, from species of trees and wildflowers dotting the pasture to species of insects, butterflies, amphibians, reptiles, birds, small mammals, etc. However, farmers face various shortcomings (constraints on using the pasture to maintain its biodiversity, reduced financial incentives, etc.).

Viscri became an attractive tourism destination after King Charles III purchased a traditional Saxon farmhouse here. He promoted the wood-pasture through his campaigning for the protection of biodiversity and environmental conservation.

Among the activities carried out by respondents during their visits to the wood-pasture, the most representative were those referring to the observation of old trees, animals, butterflies, insects and flowers, taking photographs and nature sightseeing, wagon rides in the pasture and to the sheepfold, picnics with organic products, hiking in the hills with children, attending the "mushroom hunt" event, bicycle rides, attending festivals ("Wildflowers festival"), horse riding, etc.

A very high proportion of respondents recognized the need for ecotourism development in the wood-pasture management. However, the perceived value of ecotourism in the wood-pasture is currently quite low, as it is insufficiently promoted, although it has the necessary assets to attract more nature lovers. It is also part of the Transylvanian Hills ecotourism destination.

The wood-pasture also offers many tangible benefits, both for the local community (e.g., financial, organic products, fodder and shade for animals, etc.) and for the environment (e.g., habitat for butterflies, insects, birds, reptiles, small mammals, improvement of the local microclimate, stopping soil erosion, carbon storage, etc.), which was also acknowledged by the participants in this study.

At the same time, the level of satisfaction after a wood pasture-related experience was very high and high among the interviewees, who intended to revisit the pasture or recommend it to their acquaintances.

In conclusion, the wood-pasture in Viscri is a local brand with an ancestral landscape, not only because of the centuries-old trees dotted throughout it but also due to the presence here of an ecosystem with high biocultural values whose visibility was increased considerably by King Charles III through his foundations. Although he was passionate about promoting the natural and cultural values of the grasslands, there is still a need for such initiatives, which are the responsibility of the relevant authorities.

*Future Research Directions*

Future research could focus on other wood-pastures in Transylvania that have the potential for ecotourism development. These studies would enhance their value, help protect them, and support local communities to earn additional income from such outdoor recreation activities. We are targeting the wood-pastures in Mercheașa and Ticușu villages because they also have an impressive number of centuries-old oaks (several hundred trees) compared to Viscri.

**Supplementary Materials:** The following supporting information can be downloaded at: https://www.mdpi.com/article/10.3390/f15040704/s1, File S1: Annex 1: Identifying tourists' level of awareness on the importance of practicing ecotourism on Viscri wood-pastures.

**Author Contributions:** Conceptualization, I.V., M.P. and A.N.; methodology, I.V. and A.N.; software, A.N.; validation, A.N., I.V. and A.T.; formal analysis, I.V. and M.P.; investigation, A.T.; resources, I.V., A.N. and A.T.; data curation, M.P.; writing—original draft preparation, I.V. and A.N.; writing—review and editing, I.V., A.N., A.T. and M.P.; visualization, A.T. and M.P.; supervision, I.V. and A.T.; project administration, I.V.; funding acquisition, I.V. All authors contributed equally to this work. All authors have read and agreed to the published version of the manuscript.

**Funding:** The publication of this research has been partially funded by the University of Bucharest, Romania.

**Institutional Review Board Statement:** The study was conducted according to the guidelines of the Declaration of Helsinki and approved by the Institutional Review Board (or Ethics Committee) of Bucharest University (document no. 1/3 January 2024).

**Informed Consent Statement:** Informed consent was obtained from all subjects involved in the study.

**Data Availability Statement:** The data presented in this study are available on request from the corresponding author.

**Acknowledgments:** The authors would like to thank Roxana Cuculici (Department of Regional Geography and Environment, University of Bucharest, Romania) for her valuable input and recommendations in constructing the cartographic material of this article. The authors would also like to thank the editors and anonymous reviewers for their constructive comments and helpful suggestions.

**Conflicts of Interest:** The authors declare no conflict of interest.

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
