# Peer review of "Opportunities to Capitalize on Transylvanian Wood Pastures through Nature-Based Tourism: A Case Study of Viscri Village, Brașov County, Romania"

_forests, doi:10.3390/f15040704_

Round 1

Reviewer 1 Report

Comments and Suggestions for Authors

Dear Authors,

I read your manuscript with interest. Its results shed light on both biodiversity conservation and the development of sustainable ecotourism in one of the rural areas of Transylvania. Without a doubt, its results will be of interest to developers of these scientific and applied areas of economic activity not only in Romania, but also in other countries with similar natural conditions transformed by human. At the same time, it seemed to me that some important aspects in the manuscript were not disclosed properly. They are as follows:

Main comments:

/1/ The Saxon type of farming in rural Transylvania can hardly be considered the best, if only because you did not compare it with other types, which are also characterized by the desire to preserve biodiversity and rational use of natural resources. There are quite a few such types in Europe, for example in Switzerland. It would be interesting for the reader to get at least some comparison between them. What are the advantages of the Saxon type compared to them? This should be part of the discussion. Otherwise, it looks like you're simply forcing the exclusivity of Saxon-style farming onto the reader. However, this is not obvious. More alternatives in your research!

/2/ The environmental activities of King Charles III are well known. However, you have considered the example of an area where the king has residential properties, which may additionally be a reason for attracting tourists to this area of Transylvania. However, what about other areas that lack that extra “attraction”? Are they also attractive as Viscri village? Is there an alternative to this? The broader discussion should be in the manuscript.

/3/ Are there alternative programs for the development of wood pastures in Romania, and in particular in Transylvania, that are different from yours? What are their advantages and disadvantages relative to yours? The reader should be able to see these issues from different perspectives. Moreover, this applies to readers from different countries of the world.

Minor comments:

(a)    I propose to insert the name of the country (Romania) into the title of the manuscript so that the reader can better navigate the region of study - ... Case study: Viscri village, BraÈ™ov county, Romania. Moreover, even in the Abstract there is no mention of this country.

(b)    I recommend making Figures 5 through 10 more readable by using a black font instead of a light gray one.

Therefore, I recommend moderate revision of the manuscript.

Comments on the Quality of English Language

Minor editing of English language required

Author Response

Review 1

Comments and Suggestions for Authors

Dear Authors,

I read your manuscript with interest. Its results shed light on both biodiversity conservation and the development of sustainable ecotourism in one of the rural areas of Transylvania. Without a doubt, its results will be of interest to developers of these scientific and applied areas of economic activity not only in Romania, but also in other countries with similar natural conditions transformed by human. At the same time, it seemed to me that some important aspects in the manuscript were not disclosed properly. They are as follows:

Main comments:

/1/ The Saxon type of farming in rural Transylvania can hardly be considered the best, if only because you did not compare it with other types, which are also characterized by the desire to preserve biodiversity and rational use of natural resources. There are quite a few such types in Europe, for example in Switzerland. It would be interesting for the reader to get at least some comparison between them. What are the advantages of the Saxon type compared to them? This should be part of the discussion. Otherwise, it looks like you're simply forcing the exclusivity of Saxon-style farming onto the reader. However, this is not obvious. More alternatives in your research!

R: Thank you for your valuable suggestions. We have added the following paragraphs:

As these research results have shown, the wood pasture in Viscri, managed communally according to the Saxon model, proved to be worthy of being followed by managers of other pastures of this type from Romania, despite the abovementioned shortcomings related to the insufficient subsidies provided by the state. Their advantages come from the perspective of limited use of natural resources over time because of the subsistence agricultural activities, which have reduced pressure on soil, flora and local fauna. (line 706-713)

Meanwhile, in other European regions, the pasture management system is more efficient, especially because farmers are more aware of the importance of biodiversity conservation and, hence, willing to get involved in a more sustainable use of resources. For example, Switzerland applies the silvopastoral ecosystem-oriented exploitation and conservation model that prioritises ecological diversity [112].

Spatial transformation in wood pasture landscape is determined by socioeconomic opportunities and limitations in agriculture and forest policy [113-114].

Therefore, the community use of wood pastures is a common practice in the areas where they have been preserved while the conservation regulations were introduced quite recently, in the context of sustainable development. In terms of tourist use, not all these landscapes benefitted from promotion. However, tourism is a viable alternative (also from an ecological point of view), bringing significant contributions to the community, as it happens in Viscri. (line 722-734)

/2/ The environmental activities of King Charles III are well known. However, you have considered the example of an area where the king has residential properties, which may additionally be a reason for attracting tourists to this area of Transylvania. However, what about other areas that lack that extra “attraction”? Are they also attractive as Viscri village? Is there an alternative to this? The broader discussion should be in the manuscript.

/3/ Are there alternative programs for the development of wood pastures in Romania, and in particular in Transylvania, that are different from yours? What are their advantages and disadvantages relative to yours? The reader should be able to see these issues from different perspectives. Moreover, this applies to readers from different countries of the world.

R: We have added the following paragraphs:

There are many wood pastures in Transylvania that are not well managed and preserved by the local community (e.g. Rupea, Daia, Saschiz, etc.) which completely lack any kind of promotion, much less the “additional” one carried out by the British King Charles III. Therefore, most of them are not as sought after as the one in Viscri. (line 819-822)

However, there are two other wood pastures in Transylvania whose promotion and capitalisation of ecotourism are managed by local NGOs: the one in TicuÈ™u (cycling routes, birdwatching, botanical tours, cart rides) and the one in MercheaÈ™a (walking and cycling routes, marathons). Their programmes are similar to those in Viscri, aiming at biodiversity conservation, replanting trees on pasture, pasture maintenance and promotion of ecotourism activities, even if they do not benefit from the “royal” brand. (line 823--828)

Minor comments:

  • I propose to insert the name of the country (Romania) into the title of the manuscript so that the reader can better navigate the region of study - ... Case study: Viscri village, BraÈ™ov county, Romania. Moreover, even in the Abstract there is no mention of this country.

R: Thank you for the observation. We have added it both in the title and in the abstract

  • I recommend making Figures 5 through 10 more readable by using a black font instead of a light gray one….

R: We revised the figures as suggested.

Therefore, I recommend moderate revision of the manuscript.

Comments on the Quality of English Language: Minor editing of English language required

Reviewer 2 Report

Comments and Suggestions for Authors

Thank you very much for the opportunity to review the article "Opportunities of capitalising on Transylvanian wood pastures through nature-based tourism. Case study: Viscri village, Brașov county".

I provide some comments on the paper below.

1. Please change the keywords to avoid repetition with the title of the paper. 

2. The introduction is very extensive. It presents the background of the problem of forest pastures in great detail. I, however, would perhaps concentrate on listing some examples from Europe and the problem of forest pastures in Romania. Absolutely not a mistake. I leave the decision to the chief editor. 

3. "2. The study area" - under number 2 should be the methodology, according to the editorial requirements of the journal. The numbering should be corrected as follows:

2. Methodology

2.1. The study area

3. Results and so on... 

4. I would very much like to ask you to remove or move Figure 4 e.g. to discussion, because in the results section you present only your own results. 

5. L. 422-424 You should not include this type of information in the results. The opinions of those taking part in the survey were supposed to be worked out statistically and not quoted verbatim. The same is true in L. 438-440, which is not a result.

Author Response

Review 2

Comments and Suggestions for Authors

Thank you very much for the opportunity to review the article "Opportunities of capitalising on Transylvanian wood pastures through nature-based tourism. Case study: Viscri village, Brașov county".

I provide some comments on the paper below.

  1. Please change the keywords to avoid repetition with the title of the paper. 

R: We have changed the keywords present in the title.

  1. The introduction is very extensive. It presents the background of the problem of forest pastures in great detail.

R: Thank you for your observation. We carefully considered it, and while we understand the importance of conciseness in scientific writing, we believe that the current length of the introduction is necessary to adequately contextualise the research and provide a comprehensive overview of the background and significance of the study. The introduction serves as the foundation upon which the rest of the manuscript is built, and we believe that a thorough explanation of the research context is essential for readers to fully grasp the significance of the work. Moreover, the inclusion of additional details and background information enhances the accessibility of the manuscript to a wider audience, including those who may be less familiar with the topic. At the same time, the editor confirmed to us that the length of the introduction chapter is alright.

  1. I, however, would perhaps concentrate on listing some examples from Europe and the problem of forest pastures in Romania. Absolutely not a mistake. I leave the decision to the chief editor. 

R: We have introduced other examples from Europe (line 706-733; 819-828)

  1. "2. The study area" - under number 2 should be the methodology, according to the editorial requirements of the journal. The numbering should be corrected as follows:
  2. Methodology

2.1. The study area

R: We have made the change according to the requirements of the journal.

  1. Results and so on... 
  2. I would very much like to ask you to remove or move Figure 4 e.g. to discussion, because in the results section you present only your own results. 

R: Figure 4 presents the wood pasture from Viscri; it is the authors’ contribution and a field study result, therefore we consider it to be more suitable in the results section.

  1. L. 422-424 You should not include this type of information in the results. The opinions of those taking part in the survey were supposed to be worked out statistically and not quoted verbatim. The same is true in L. 438-440, which is not a result.

R: Thank you very much for your suggestion. The use of quotes in our qualitative analysis was motivated by a commitment to preserving the authenticity of participants' voices, and promoting transparency in our research process. We believe this approach enhances our data's credibility and richness and contributes to a more nuanced understanding of the phenomena under investigation. We appreciate your attention to this aspect of our manuscript and welcome any further feedback or suggestions you may have.

Reviewer 3 Report

Comments and Suggestions for Authors

This paper is well-organized article. Because I read this paper is smoothly and easy to follow without any problem, it can make readers know what the topic they want to provide.

 The section of introduction has around 6 pages. There are 3 sub-sections included. It provides enough background and problem’s descriptions. I think this part should be check, because too many information from the literature is not necessary.

 The section of the study is clearly and easy to understand, the Figure 1 based on the GIS can show a overview for readers to take a view quickly.

 The section of Methodology is clearly and no problem. It is a survey through questionnaire, then social network analysis and statistical frequency analysis are used for data analysis.

 The section of Results is clearly and fruitful. Some figures are beautiful, but Figure 10 is not good since the color if printout will difficult to look. I think that the color should be changed. And some sub-sections to third level, it is not good. For example, 4.2.3. Prospects for the capitalisation of wood pastures.

 The section of Discussions

5.1. Viscri wood pasture – a brief analysis (line 606), the title is not good here, because the Discussion is not a section of analysis.

 Figure 11 can turn another side??  It is good, but difficult to read.

 The section of Conclusions is simple but strongly to make a summary.

Author Response

Review 3

Comments and Suggestions for Authors

This paper is well-organized article. Because I read this paper is smoothly and easy to follow without any problem, it can make readers know what the topic they want to provide.

 The section of introduction has around 6 pages. There are 3 sub-sections included. It provides enough background and problem’s descriptions. I think this part should be check, because too many information from the literature is not necessary.

R: Thank you for your observation. We carefully considered it, and while we understand the importance of conciseness in scientific writing, we believe that the current length of the introduction is necessary to adequately contextualise the research and provide a comprehensive overview of the background and significance of the study. The introduction serves as the foundation upon which the rest of the manuscript is built, and we believe that a thorough explanation of the research context is essential for readers to fully grasp the significance of the work. Moreover, the inclusion of additional details and background information enhances the accessibility of the manuscript to a wider audience, including those who may be less familiar with the topic. At the same time, the editor confirmed to us that the length of the introduction chapter is alright.

The section of the study is clearly and easy to understand, the Figure 1 based on the GIS can show a overview for readers to take a view quickly.

 The section of Methodology is clearly and no problem. It is a survey through questionnaire, then social network analysis and statistical frequency analysis are used for data analysis.

 The section of Results is clearly and fruitful. Some figures are beautiful, but Figure 10 is not good since the color if printout will difficult to look. I think that the color should be changed.

R: We revised the figures as suggested (see the manuscript).

And some sub-sections to third level, it is not good. For example, 4.2.3. Prospects for the capitalisation of wood pastures….

R: We deleted the third-level sub-sections.

The section of Discussions

5.1. Viscri wood pasture – a brief analysis (line 606), the title is not good here, because the Discussion is not a section of analysis….

R: Thank you for your observation, it was changed in: Viscri wood pasture – a brief overview.

Figure 11 can turn another side??  It is good, but difficult to read

R: We will ask the editors if inserting it as landscape on a separate page is possible.

The section of Conclusions is simple but strongly to make a summary.

Round 2

Reviewer 1 Report

Comments and Suggestions for Authors

The authors responded to my comments and made changes to the manuscript.

Comments on the Quality of English Language

Minor editing of English language required

Author Response

Comments and Suggestions for Authors:

The authors responded to my comments and made changes to the manuscript.

Comments on the Quality of English Language: Minor editing of English language required

R: Thank you for the remarks. Minor editing of English has been provided.